# Ligand activation mechanisms of human KCNQ2 channel

Demin Ma [1,2,13], Yueming Zheng[3,4,13], Xiaoxiao Li[1,2,13], Xiaoyu Zhou[3,4,13], Zhenni Yang[1,2], Yan Zhang[1,2], Long Wang[3], Wenbo Zhang[3], Jiajia Fang[5], Guohua Zhao[5], Panpan Hou[6], Fajun Nan[3,4], Wei Yang [1], Nannan Su [1], Zhaobing Gao [3,4,7] ✉ & Jiangtao Guo [1,2,8,9,10,11,12] ✉

The human voltage-gated potassium channel KCNQ2/KCNQ3 carries the neuronal M-current, which helps to stabilize the membrane potential. KCNQ2 can be activated by analgesics and antiepileptic drugs but their activation mechanisms remain unclear. Here we report cryo-electron microscopy (cryo-EM) structures of human KCNQ2-CaM in complex with three activators, namely the antiepileptic drug cannabidiol (CBD), the lipid phosphatidylinositol 4,5-bisphosphate (PIP₂), and HN37 (pynegabine), an antiepileptic drug in the clinical trial, in an either closed or open conformation. The activator-bound structures, along with electrophysiology analyses, reveal the binding modes of two CBD, one PIP₂, and two HN37 molecules in each KCNQ2 subunit, and elucidate their activation mechanisms on the KCNQ2 channel. These structures may guide the development of antiepileptic drugs and analgesics that target KCNQ2.

The voltage-gated potassium channel KCNQ family contains five members, among which KCNQ1 carries the cardiac slow delayed-rectifier ($I_{Ks}$) current and repolarizes the cardiac action potential[1–3] whereas the KCNQ2-5 channels modulate the neuronal excitability[4]. Among the neuronal KCNQ channels, KCNQ2 and KCNQ3 form homo- and hetero-tetrameric channels and are responsible for the M-current, which contributes to the stability of the membrane potential[5–7]. Mutations in the KCNQ2 or KCNQ3 lead to hyper-excitation of neurons and the occurrence of seizures, such as benign familial neonatal seizures (BFNS or BFNE)[8,9] and epileptic encephalopathy (EE)[10–12], making KCNQ2 an important target of antiepileptic drugs[13–15]. Besides, modulation of KCNQ2 is a potential therapeutic strategy for the treatment of some other neuronal diseases such as pain[16–19], Parkinson's disease[20,21], and schizophrenia[22], which are also characterized by hyper-excitation of neurons.

To develop new antiepileptic drugs and analgesics, a number of natural or synthetic compounds targeting KCNQ2 have been identified. Retigabine (RTG) was an approved antiepileptic drug that potentiates the activation of potassium channels[23,24]. The non-opioid analgesic flupirtine, which has been used for the treatment of acute and chronic pain

[1]Department of Biophysics and Department of Neurology of the Fourth Affiliated Hospital, Zhejiang University School of Medicine, Hangzhou, Zhejiang 310058, China. [2]Nanhu Brain-computer Interface Institute, Hangzhou 311100, China. [3]State Key Laboratory of Drug Research, Shanghai Institute of Materia Medica, Chinese Academy of Sciences, Shanghai 201203, China. [4]University of Chinese Academy of Sciences, 19A Yuquan Road, Beijing 100049, China. [5]Department of Neurology of the Fourth Affiliated Hospital, Zhejiang University School of Medicine, Yiwu, Zhejiang 322000, China. [6]Dr. Neher's Biophysics Laboratory for Innovative Drug Discovery, State Key Laboratory of Quality Research in Chinese Medicine, Macau University of Science and Technology, Taipa, Macao SAR, China. [7]Zhongshan Institute for Drug Discovery, Shanghai Institute of Materia Medica, Chinese Academy of Sciences, Zhongshan 528437, China. [8]Department of Cardiology, Key Laboratory of Cardiovascular Intervention and Regenerative Medicine of Zhejiang Province, Sir Run Run Shaw Hospital, Zhejiang University School of Medicine, Hangzhou, Zhejiang 310016, China. [9]Liangzhu Laboratory, Zhejiang University Medical Center, 1369 West Wenyi Road, Hangzhou, Zhejiang 311121, China. [10]NHC and CAMS Key Laboratory of Medical Neurobiology, MOE Frontier Science Center for Brain Science and Brain-machine Integration, School of Brain Science and Brain Medicine, Zhejiang University, Hangzhou, China. [11]State Key Laboratory of Plant Physiology and Biochemistry, College of Life Sciences, Zhejiang University, Hangzhou, Zhejiang 310058, China. [12]Cancer Center, Zhejiang University, Hangzhou, Zhejiang 310058, China. [13]These authors contributed equally: Demin Ma, Yueming Zheng, Xiaoxiao Li, Xiaoyu Zhou. ✉e-mail: zbgao@simm.ac.cn; jiangtaoguo@zju.edu.cn

in Europe, can also activate KCNQ2[25,26]. Cannabidiol (CBD) is one of the main effective compounds in cannabis and a clinically effective anti-epileptic drug. Pharmacological and structural studies reveled that multiple ionotropic receptors contributing to the CBD's antiepileptic activity include blockade of Nav and desensitization of TRPV channels[27–30]. Recently, CBD was shown to directly activate the KCNQ2/KCNQ3 channel[31,32]. We previously developed a series of KCNQ2 activators, exemplified by ztz240 and HN37 (pynegabine)[33,34], the latter has been in the clinical trial phase I[34]. Because many of these KCNQ2 activators including RTG can cause side effects due to their poor target selectivity[35–37], revealing the structural basis for the ligand recognition and activation of KCNQ2 will advance the development of KCNQ2-selective activators and overcome the side effects of current analgesics and antiepileptic drugs.

Recently, structures of KCNQ channels revealed the overall architecture and gating mechanism of KCNQs[38–44]. We determined structures of KCNQ2 in complex with its activators RTG and ztz240, both in the closed state[38]. KCNQ2 adopts the canonical structural organization of the tetrameric $K_V$ channels in a domain-swapped arrangement[38]. Each KCNQ2 subunit contains six transmembrane segments (S1–S6): S1–S4 form the voltage-sensing domain (VSD) and the S5–S6 constitute the pore domain (PD). The C-terminal domain (CTD), which contains three helices, namely HA, HB, and HC, interacts with the calcium-modulated protein calmodulin (CaM). Upon membrane depolarization, the VSD activates and induces conformational changes to open the PD via the coupling of the VSD and the PD (electro-mechanical coupling, E–M coupling). The lipid phosphatidylinositol 4,5-bisphosphate ($PIP_2$) is required for the E–M coupling and depleting $PIP_2$ results in a closed pore with normal VSD activation[45]. Moreover, many epilepsy-associated mutations in KCNQ2 disrupt E-M coupling and RTG can restore the E–M coupling in some cases[46]. Since all the current KCNQ2 structures are in decoupled conformation with activated VSDs and a closed PD[38], how $PIP_2$ and antiepileptic drugs activate KCNQ2 by enhancing the E–M coupling remains largely unknown.

In this report, we use cryo-electron microscopy (cryo-EM) and electrophysiology to study the modulation mechanisms of KCNQ2 E−M coupling by the activators $PIP_2$, CBD, and HN37. We present 2.5–3.5 Å resolution structures of human KCNQ2-CaM complex in the CBD-bound closed, HN37-bound closed, and CBD-$PIP_2$-bound open states (Table 1). This study not only elucidates the molecular basis for the activation of the KCNQ2 by different small molecules but also may guide the development of KCNQ2-selective activators as potential analgesics and antiepileptic drugs.

## Results

### Structure determination of KCNQ2-CaM in the presence of $PIP_2$

Previously, we determined the apo KCNQ2-CaM structure (KCNQ2-CaM$_{apo}$, PDB: 7CR3)[38], which was in decoupled conformation with activated VSDs and a closed gate. Because $PIP_2$ can potentiate the activation of KCNQ channels by enhancing the E−M coupling, to capture an open-state KCNQ2, we solved the structure of the KCNQ2-CaM complex in the presence of 1 mM $PIP_2$ (KCNQ2-CaM$_{PIP2(-)}$, where (-) indicates that $PIP_2$ was added in the protein sample but not observed in the structure throughout this paper) at 2.7 Å resolution (Supplementary Fig. 1a–h, Table 1). Unfortunately, we were unable to capture the open-state KCNQ2 structure with the presence of $PIP_2$ in the sample, nor $PIP_2$ molecule was observed in the map, suggesting that $PIP_2$ does not tightly bind to KCNQ2 in vitro under our structure determination conditions. The low affinity of $PIP_2$ bound to KCNQ2 was reported in a previous electrophysiological study which showed that the half-maximal value ($EC_{50}$) for diC8-$PIP_2$ on KCNQ2 was ~205 μM[47]. By contrast, in the case of the human KCNQ1-CaM-KCNE3 complex, with the presence of $PIP_2$, the open-state KCNQ1 structure was readily obtained[41].

The KCNQ2-CaM$_{PIP2(-)}$ structure is similar to the KCNQ2-CaM$_{apo}$ structure, with activated VSDs and a closed gate[38]. In the VSD, the side chain of Arg210 (R5) points to the gating charge transfer center formed by Phe137, Glu140, and Asp172 (Supplementary Fig. 1i). The activation gate is closed by two layers of constriction formed by residues Ser314 and Leu318, with the shortest diagonal atom-to-atom distance of 4–6 Å at the activation gate (Supplementary Fig. 1j). In the CTD, HA and HB helices are wrapped by CaM. As no inter-subunit interactions are observed in the CaM, the conformation of HA/HB/CaM is likely stabilized by interactions between the linker connecting the fifth and sixth helix of CaM (H5-H6 linker) and S0 and the S2–S3 linker from the VSD (Supplementary Fig. 1k).

### Structural basis for the CBD recognition

To reveal the molecular mechanism of CBD activation on KCNQ2-CaM, we next determined the CBD-bound KCNQ2-CaM structure (KCNQ2-CaM$_{CBD}$) at 3.3 Å resolution (Fig. 1a, Table 1, Supplementary Figs. 2a–e, 3a). In the TMD, the high-quality map of KCNQ2-CaM$_{CBD}$ allows us to clearly resolve two CBD molecules located in the hydrophobic cleft formed by S6 from the first subunit (S6$_I$), S5, pore helix (PH), and S6 from the second subunit (S5$_{II}$, PH$_{II}$, and S6$_{II}$), and S1 from the third subunit (S1$_{III}$) (Fig. 1a, b). The configurations of CBDs were assigned based on their large cyclohexenyl head group and the short hydrophobic tail of the pentylphenyl group (Fig. 1b). One CBD molecule (CBD$_A$) sits deep in the classical fenestration pocket of voltage-gated ion channels lined by S6$_I$, S5$_{II}$, and S6$_{II}$ (Fig. 1c). CBD$_A$ mainly forms hydrophobic interactions with residues Leu299, Ile300 and Phe304 in S6$_I$, Leu232, Trp236, and Phe240 in S5$_{II}$, and Phe305, Pro308, and Leu312 in S6$_{II}$ (Fig. 1c). In addition, it also forms hydrogen bonds with main-chain carbonyls of Leu299 in S6$_I$ and Leu232 in S5$_{II}$ (Fig. 1c). The other CBD molecule (CBD$_B$) is localized at an upper, shallow pocket formed by S5$_{II}$, PH$_{II}$, and S1$_{III}$, covering the upper part of CBD$_A$ (Fig. 1d). Similarly, CBD$_B$ also mainly forms hydrophobic interactions with surrounding residues such as Leu299 and Ile300 in S6$_I$, Trp236, Phe240,

## Table 1 | Summary of the KCNQ2-CaM structures reported in this study

| Structure | Ligand in the sample | Ligand in the structure | Gate | Resolution |
|---|---|---|---|---|
| KCNQ2-CaM$_{PIP2(-)}$[a] | $PIP_2$ | none | closed | 2.7 Å |
| KCNQ2-CaM$_{CBD}$ | CBD | two CBDs | closed | 3.0 Å |
| KCNQ2-CaM$_{CBD-PIP2}$[b] | $PIP_2$, CBD | one $PIP_2$, two CBDs | open | 3.1 Å |
| KCNQ2-CaM$_{F104A-CBD-PIP2(-)-I}$ | $PIP_2$, CBD | one CBD | closed | 2.7 Å |
| KCNQ2-CaM$_{F104A-CBD-PIP2-II}$ | $PIP_2$, CBD | one $PIP_2$, one CBD | open | 3.5 Å |
| KCNQ2-CaM$_{HN37}$ | HN37 | two HN37 | closed | 2.5 Å |
| KCNQ2-CaM$_{HN37-PIP2(-)}$ | $PIP_2$, HN37 | two HN37 | closed | 2.7 Å |
| KCNQ2-CaM$_{PIP2-HN37}$ | $PIP_2$, HN37 | one $PIP_2$, one HN37 | open | 3.3 Å |

[a]The subscripted (-) after $PIP_2$ in all structures indicates that $PIP_2$ was added into the samples but not resolved in the structures.
[b]The orders of the ligands in the subscripts of all structures reflect sequences of these ligands added into the protein samples.

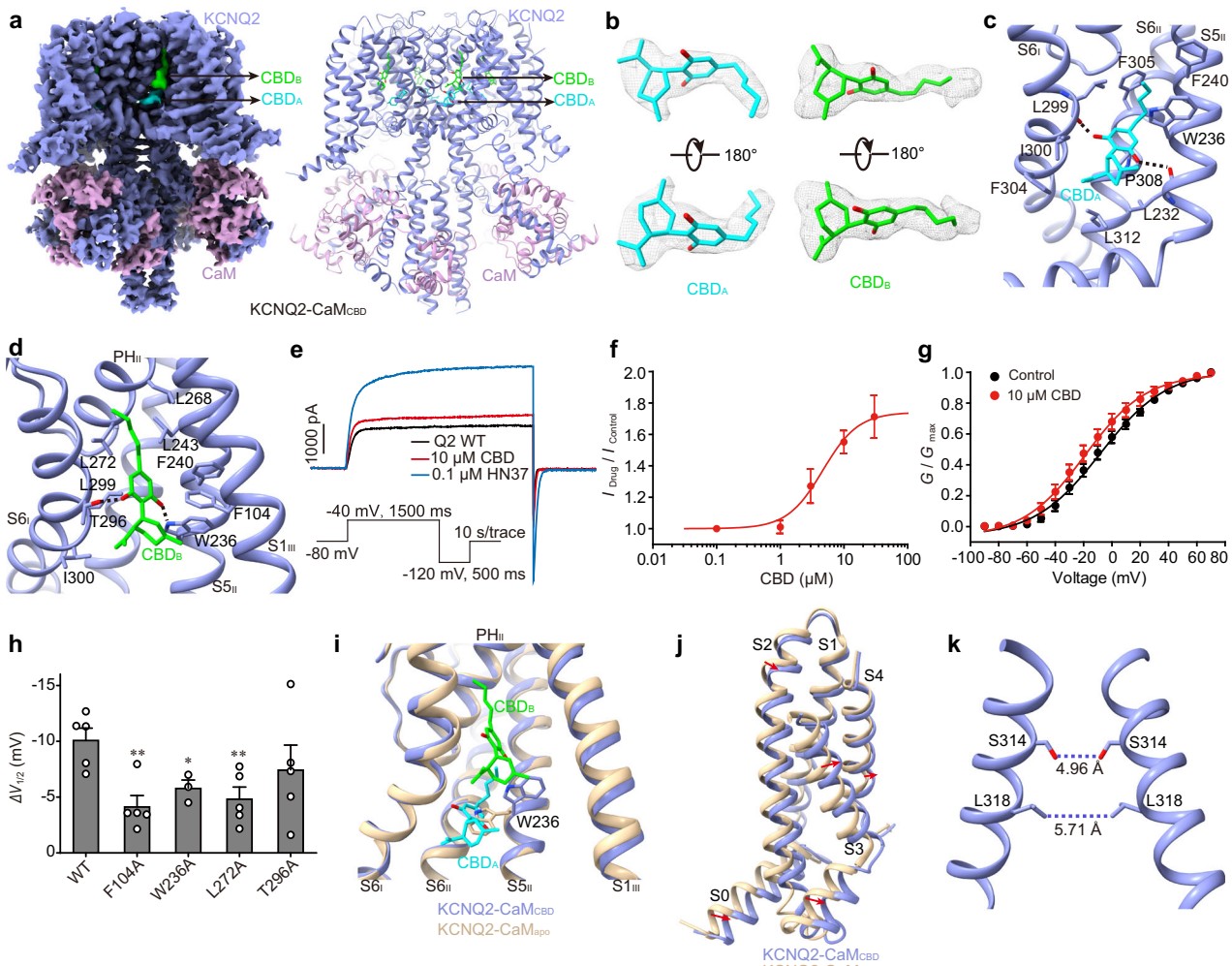

**Fig. 1 | The CBD-bound KCNQ2-CaM structure (KCNQ2-CaM$_{CBD}$). a** The 3D reconstruction and the cartoon model of KCNQ2-CaM$_{CBD}$. **b** The density maps of two CBD molecules in different orientations at the contour level of 3.6 σ. **c** Interactions between CBD$_A$ and KCNQ2. The dashed lines indicate hydrogen bonds. **d** Interactions between CBD$_B$ and KCNQ2. The dashed lines indicate hydrogen bonds. **e** Representative traces of KCNQ2 currents in the absence (black) and presence of 10 μM CBD (red) or 100 nM HN37 (blue). (Inset) The recording protocol. The cell was clamped at −80 mV and the KCNQ2 currents were elicited by a depolarizing voltage step of −40 mV for 1500 ms, followed by a hyperpolarizing step to −120 mV for 500 ms. **f** The dose-response curve of CBD on the current amplitude of KCNQ2 at −40 mV. Data are presented as mean ± SEM; $n = 3$ (0.1 μM), 3 (1 μM), 5 (3 μM), 5 (10 μM), and 3 (30 μM) individual cells. **g** Voltage-dependent activation curves of KCNQ2 before (black) and after (red) application of 10 μM CBD. Data are presented as mean ± SEM; $n = 5$ individual cells. **h** The half maximal activation voltage shift ($\Delta V_{1/2}$) of WT KCNQ2 and mutants by 10 μM CBD. Data are presented as mean ± SEM. An unpaired two-tailed $t$ test was performed to compare WT and each mutant. $**p = 0.0036$ for F104A, $*p = 0.0307$ for W236A, $**p = 0.0083$ for L272A, and $p = 0.3123$ for T296A. For W236A, $n = 3$ individual cells; For WT and other mutants, $n = 5$ individual cells. **i** CBDs induce a rotamer change of Trp236 in S5. **j** Structural comparison of VSDs from KCNQ2-CaM$_{apo}$ and KCNQ2-CaM$_{CBD}$ when the whole channels are aligned. **k** The closed activation gate of KCNQ2-CaM$_{CBD}$. The dashed lines show diagonal atom-to-atom distance (in Å) at the constriction-lining residues Ser314 and Leu318. For (**f**–**h**), source data are provided in the Source Data file.

and Leu243 in S5$_{II}$, Leu268 and Leu272 in PH$_{II}$, and Phe104 in S1$_{III}$ (Fig. 1d). Specifically, CBD$_B$ forms a hydrogen bond with the side chains of the conserved Trp236 in S5$_{II}$, as well as Thr296 in S6$_I$ (Fig. 1d).

To confirm the two CBD binding sites revealed by the structure data, we expressed KCNQ2 in Chinese hamster ovary (CHO) cells and recorded the current using the whole-cell patch-clamp technique (Fig. 1e). Under −40 mV, 10 μM CBD enhances the outward current of KCNQ2 by an average factor of 1.55 ± 0.07 (Fig. 1f). CBD enlarges the current amplitudes in a dose-dependent manner with an EC$_{50}$ value of 4.50 ± 1.79 μM ($n = 5$) at −40 mV (Fig. 1f). 10 μM CBD also potentiates the voltage activation of KCNQ2 by left-shifting the conductance-voltage ($G–V$) curve, with an average half maximal activation voltage shift ($\Delta V_{1/2}$) of −10.11 ± 1.08 mV (Fig. 1g). Thus, CBD increases the voltage sensitivity of KCNQ2, similar to the previous report[31]. We next examined the CBD potentiation effect on KCNQ2 channels with mutations on the CBD-interacting residues. Among the nine mutants

we tested, five (L232A, L268A, A295L, F304A, and F305A) were non-functional and one (T296A) showed similar CBD sensitivity as WT (Fig. 1h, Supplementary Fig. 4, Supplementary Data 1). The rest three, namely F104A, W236A, and L272A, all displayed a significantly reduced potentiation activity of CBD by attenuating the left shift of the $G–V$ curve ($\Delta V_{1/2}$) (Fig. 1h, Supplementary Fig. 4). While Phe104 and Leu272 specifically interact with CBD$_B$, Trp236 forms interactions with both CBD molecules. These results collectively support the binding sites of CBDs in KCNQ2.

Structural comparison of KCNQ2-CaM$_{CBD}$ and KCNQ2-CaM$_{apo}$ reveals that the two CBDs induce a rotamer change of Trp236 in S5$_{II}$, which eliminates the potential clash with CBD$_A$ on one hand and facilitates the formation of a hydrogen bond with CBD$_B$ on the other hand (Fig. 1i). In addition, a 2–3 Å shift of the whole VSD is observed (Fig. 1j). However, this disturbance of VSD does not completely destroy interactions between VSD and the attached CaM (Fig. 1a). The gate in

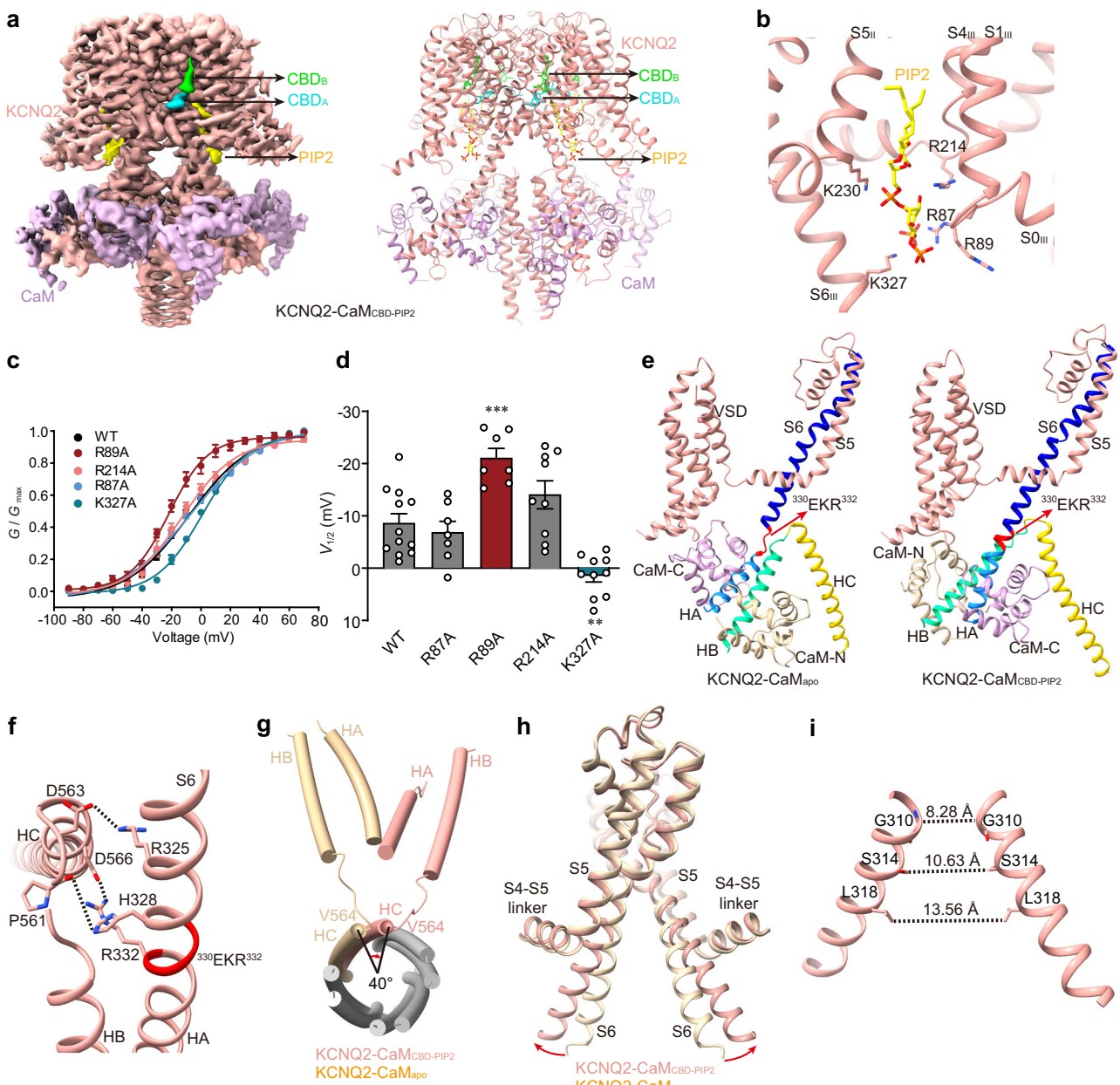

**Fig. 2 | The PIP₂- and CBD-bound KCNQ2-CaM structure (KCNQ2-CaM_CBD-PIP2).** **a** The 3D reconstruction and the cartoon model of KCNQ2-CaM_CBD-PIP2. **b** The binding site of PIP₂ in KCNQ2. Side chains of residues in KCNQ2 involved in the interactions with PIP₂ are shown as sticks. **c** Voltage-dependent activation curves of WT KCNQ2 and mutants. Data are presented as mean ± SEM. *n* values are indicated in (**d**). **d** $V_{1/2}$ values of WT KCNQ2 and mutants. Data are presented as mean ± SEM. One-way ANOVA with Dunnett's multiple comparisons test was applied. *p* and *n* values compared to WT (*n* = 12) are *p* = 0.9379 and *n* = 7 for R87A, ***p* = 0.0003 and *n* = 7 for R89A, *p* = 0.1484 and *n* = 9 for R214A, and ***p* = 0.0015 and *n* = 10 for R327A. *n* represents the number of experiments from individual cells. **e** Different structure arrangements of CTD and CaM in KCNQ2-CaM_apo and KCNQ2-CaM_CBD-PIP2. S6, S6-HA linker, HA, HB, and HC helices of KCNQ2 are colored in blue, red, cornflower-blue, green, and yellow, respectively. The N-lobe and C-lobe of CaM are shown in wheat and pink, respectively. **f** Interactions between S6 and HB-HC linker. The dashed lines indicate the salt bridge Arg332-Asp566 and the hydrogen bonds Arg325-Asp563 and His328-Pro561. **g** Structural comparison of the HA, HB, and HC helices in KCNQ2-CaM_apo (wheat) and KCNQ2-CaM_CBD-PIP2 (salmon) in the top view with TMD and HA/HB from the other three subunits omitted for clarity. The N-terminal residue Val564 of HC rotates by ~40° from KCNQ2-CaM_apo to KCNQ2-CaM_CBD-PIP2. **h** Overlay of the pore domains of KCNQ2-CaM_apo (wheat) and KCNQ2-CaM_CBD-PIP2 (salmon) structures showing the conformational change in the ion-conducting pathway. The S6 helix bends outward upon PIP₂ binding at the point of PAG segment. **i** The open activation gate of KCNQ2-CaM_CBD-PIP2. The dashed lines show diagonal atom-to-atom distance (in Å) at the constriction-lining residues Gly310, Ser314, and Leu318. For (**c**, **d**), source data are included in the Source Data file.

KCNQ2-CaM_CBD remains closed in the absence of PIP₂ (Fig. 1k), hindering the elucidation of the activation mechanism of CBD.

## The CBD- and PIP₂-bound open-state structure of KCNQ2-CaM

To decipher the CBD and PIP₂ activation mechanism, we then determined the PIP₂-CBD-bound KCNQ2-CaM structure (KCNQ2-CaM_CBD-PIP2)

at 3.1 Å resolution (Fig. 2a, Table 1, Supplementary Figs. 2f–i, 3b). Like those in KCNQ2-CaM_CBD, two CBD molecules are clearly observed in the map of KCNQ2-CaM_CBD-PIP2 (Fig. 2a). In addition, between VSD and PD, one PIP₂ molecule is assigned, with its fatty acid chains switched by S1_III/S4_III and S5_II in the transmembrane region and its inositol 1,4,5-trisphosphate head group pointing to S6_III helix in the cytosol (Fig. 2b).

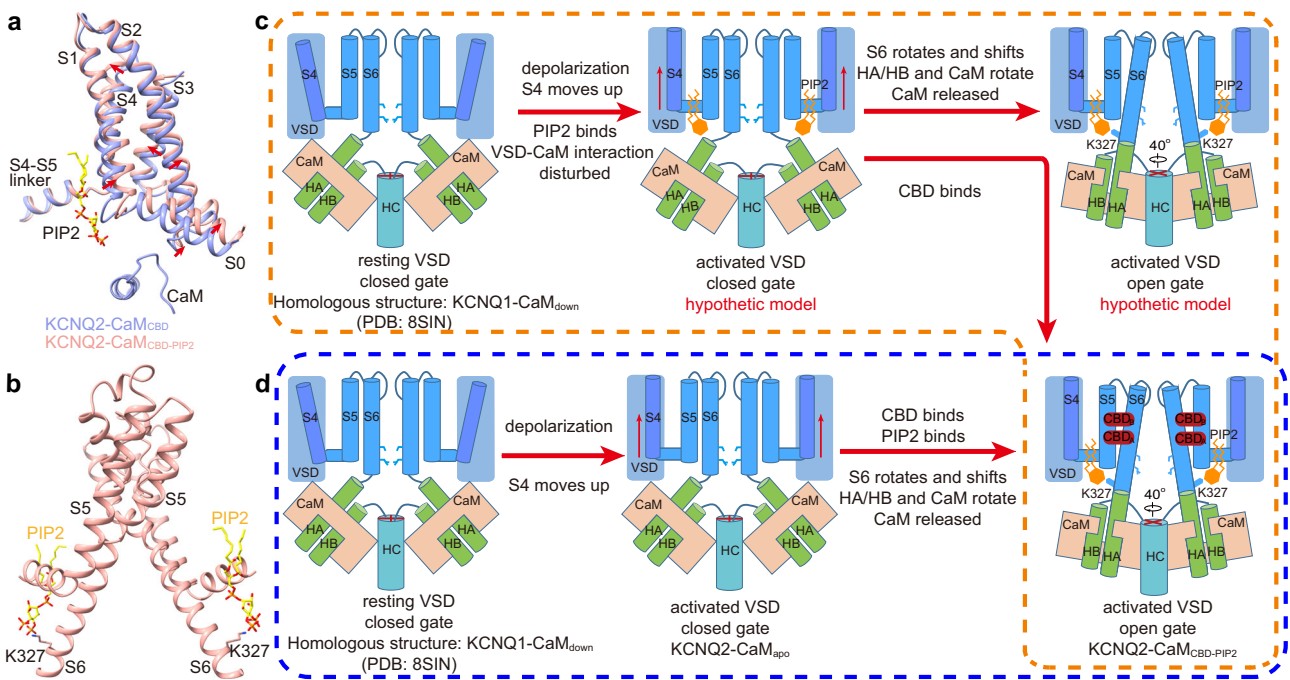

**Fig. 3 | The CBD and PIP2 activation mechanism. a** Structural comparison of VSDs from KCNQ2-CaM$_{CBD}$ and KCNQ2-CaM$_{CBD-PIP2}$ when the whole channels are aligned. **b** PIP$_2$ "pulls" S6 up by interaction with the Lys327 in S6. **c** The proposed CBD and PIP$_2$ activation mechanism of KCNQ2 in vivo. **d** The proposed CBD and PIP$_2$ activation model of KCNQ2 in vitro.

The inositol 1,4,5-trisphosphate group interacts with positively charged residues Arg87 in S0$_{III}$, Arg89 in S1$_{III}$, Arg214 in S4$_{III}$, Lys230 in S5$_{II}$, and Lys327 in S6$_{III}$ (Fig. 2b). This PIP$_2$ binding site is similar to the P2 site of PIP$_2$ in the structure of KCNQ4 in complex with PIP$_2$ and the agonist ML213[43]. Sequence alignment of KCNQ1-5 show that these PIP$_2$-interacting residues are highly conserved across KCNQ2-5 but not in KCNQ1 (Supplementary Fig. 5), which may explain why this PIP$_2$-binding site is not observed in KCNQ1 structures[39,41].

To test whether these five positively charged residues are essential for the voltage activation of KCNQ2 via the PIP$_2$, we then introduced mutations at the PIP$_2$ binding site in KCNQ2 and recorded the currents of CHO cells expressing WT and mutant KCNQ2. While the mutations R87A and R214A did not change the voltage activation of KCNQ2, K327A reduced the voltage sensitivity of KCNQ2 by right-shifting the $G-V$ curve (Fig. 2c, d). In addition, the mutant K230A lost the activity, emphasizing its essential role in the PIP$_2$ binding. The mutation R89A left-shifts the $G-V$ curve, whose mechanism remains unknown (Fig. 2c, d).

The binding of CBD and PIP$_2$ induces a large structural arrangement of the CTD and CaM, as well as S6 (Fig. 2e–i). HA and HB, together with their surrounding CaM, rotate by almost 180°. HB which sits on the right side of HA in KCNQ2-CaM$_{apo}$ moves to the left side of HA in KCNQ2-CaM$_{CBD-PIP2}$ (Fig. 2e). Along with this rotation, S6 and HA, which were connected by a loop linker of residues $^{330}$EKR$^{332}$, now form a continuous straight helix (Fig. 2e, Supplementary Movies 1 and 2). This re-folding of the $^{330}$EKR$^{332}$ motif is likely energy-favorable, as multiple interactions including the salt bridge and hydrogen bonds are introduced at the interface of S6/HA and the HB-HC linker (Fig. 2f). Consequently, the coiled-coil-forming helix HC, which is directly connected to HB, rotates by ~40° (Fig. 2g). This structural arrangement of CTD and S6 causes the bending of the S6 C-terminal half at the conserved "PAG" segment away from the central axis of the channel pore, resulting in the opening of the activation gate with the shortest diagonal atom-to-atom distance extending to 8–9 Å at the activation gate (Fig. 2h, i, and

Supplementary Movie 3). Therefore, bound with both CBD and PIP$_2$, KCNQ2-CaM$_{CBD-PIP2}$ is in an open state. This KCNQ2-CaM$_{CBD-PIP2}$ structure resembles the previously reported PIP$_2$-bound open-state structures of KCNQ1 and KCNQ4[39,41,43].

## The CBD and PIP$_2$ activation mechanism
How do CBD and PIP$_2$ potentiate the voltage activation of KCNQ2? First, the structure alignment of KCNQ2-CaM$_{apo}$ and KCNQ2-CaM$_{CBD}$ shows that the CBD induces a 2–3 Å shift of VSD, which tends to disturb interactions of VSD and CaM (Fig. 1j). Second, structure alignment KCNQ2-CaM$_{CBD}$ and KCNQ2-CaM$_{CBD-PIP2}$ shows that PIP$_2$ induces a further 3–4 Å shift of VSD (Fig. 3a), which completely destroys interactions of VSD and CaM, resulting in the release of CaM and HA/HB from the VSD, followed by the rotation of CaM and HA/HB, as well as the folding of S6 and HA into one continuous helix (Fig. 2d). In addition, by interaction with the Lys327 in S6, PIP$_2$ "pulls" S6 up and stabilizes the open-state conformation (Fig. 3b).

These CBD and PIP$_2$ activation insights are obtained from comparisons of the KCNQ2-CaM structures with or without ligands. In fact, in vivo the channel activation is initiated by the depolarization of membrane potential instead of the binding of PIP$_2$ because the PIP$_2$ is already bound in KCNQ2. Then what the activation process is in vivo? Based on the above analyses, we propose that in the absence of CBD, upon membrane potential depolarization, S4 moves up, which would change the conformation and (or) the location of PIP$_2$. PIP$_2$ then disturbs interactions between VSD and CaM, resulting in the release of CaM and HA/HB from the VSD on one hand, and pulls S6 up to stabilize the open-state conformation on the other hand (Fig. 3c). However, in vitro the binding of PIP$_2$ in the activated VSD is not that strong and PIP$_2$ is easy to dissociate, which can explain why we were unable to capture the PIP$_2$-bound open-state KCNQ2-CaM structure in the presence of PIP$_2$ alone (KCNQ2-CaM$_{PIP2(-)}$) (Fig. 3d). The binding of CBDs likely stabilizes the PIP$_2$-bound state by pushing VSD away to ensure a sufficient hydrophobic pocket between VSD and PD for PIP$_2$, enabling us to capture the CBD-PIP$_2$-bound open-state structure (Fig. 3d).

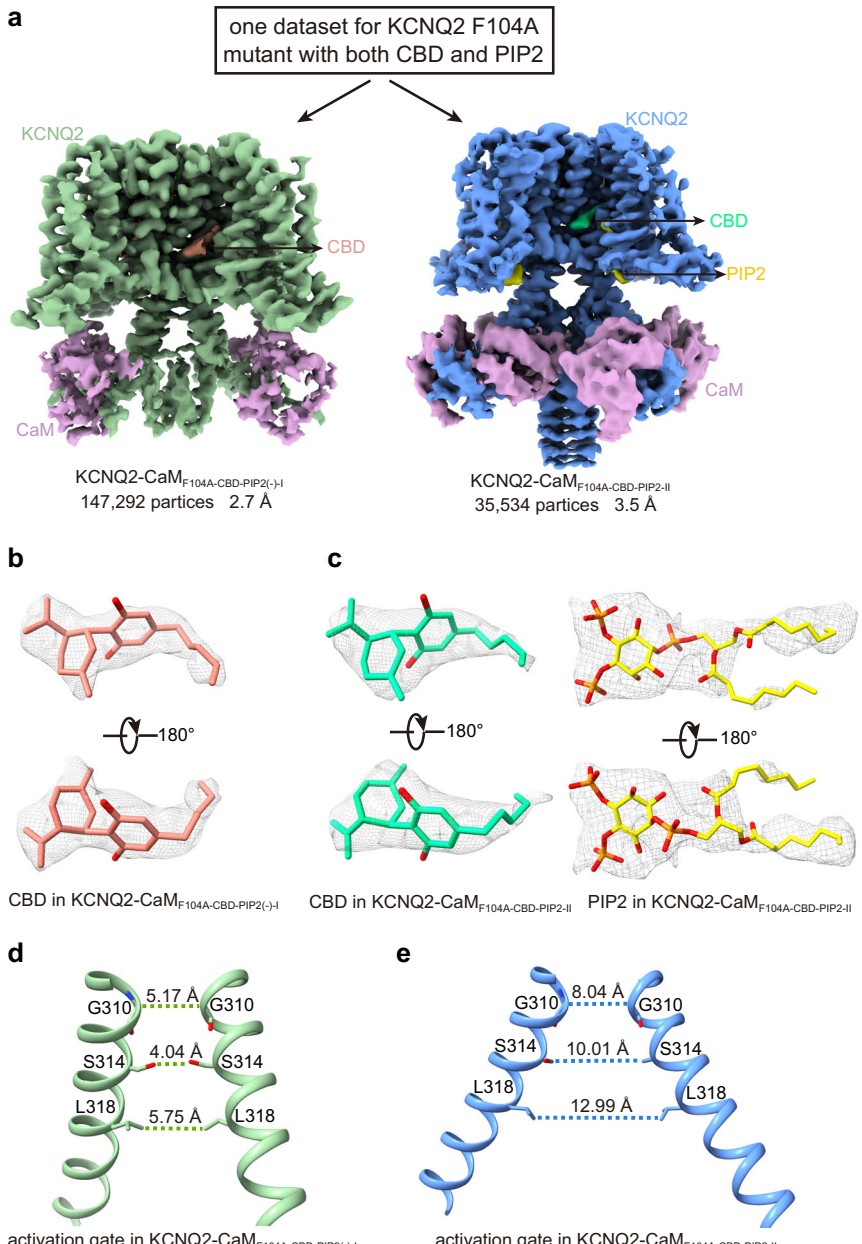

**Fig. 4 | Two structures of the KCNQ2-CaM F104A mutant determined from one dataset in the presence of CBD and PIP₂. a** Two different KCNQ2-CaM$_{F104A}$ maps were reconstructed from one dataset for KCNQ2 F104A mutant with both CBD and PIP₂. **b** The density maps of one CBD molecule in different orientations at the contour level of 3.7 σ. **c** The density maps of one CBD molecule and one PIP₂ molecule in different orientations at the contour level of 3.7 σ and 3.5 σ, respectively. **d** The closed activation gate of KCNQ2-CaM$_{F104A-CBD-PIP2(-)-I}$. The dashed lines show diagonal atom-to-atom distance (in Å) at the constriction-lining residues Gly310, Ser314, and Leu318. **e** The open activation gate of KCNQ2-CaM$_{F104A-CBD-PIP2-II}$. The dashed lines show diagonal atom-to-atom distance (in Å) at the constriction-lining residues Gly310, Ser314, and Leu318.

Therefore, both PIP₂ and CBD potentiate the voltage activation of KCNQ2 by enhancing the E–M coupling.

## Different roles of the two CBDs

While CBD$_A$ sits deep in the classical fenestration pocket and coincides with the RTG in the structure of KCNQ2[38], CBD$_B$ is localized at the pocket surface and binds to KCNQ2 relatively loosely (Fig. 1a, i). We then ask whether these two sites are both essential for the potentiation of the KCNQ2 voltage activation. To probe the role of CBD$_B$, we introduced an alanine substitution on the residue Phe104 in S1, which directly interacts with CBD$_B$. Electrophysiological studies show that the activation of CBD is largely affected in the mutant F104A (Fig. 1h). We next collected cryo-EM data using the F104A mutant sample

incubated with both PIP₂ and CBD. Interestingly, two different KCNQ2-CaM$_{F104A}$ maps were reconstructed from one dataset, which were designated KCNQ2-CaM$_{F104A-CBD-PIP2(-)-I}$ and KCNQ2-CaM$_{F104A-CBD-PIP2-II}$ at 2.7 Å and 3.5 Å resolutions, respectively (Fig. 4a, Table 1, Supplementary Figs. 2j–q, 3c, d). In both maps, CBD$_A$ molecules maintain strong densities but those for CBD$_B$ are barely observed (Fig. 4a–c). Interestingly, KCNQ2-CaM$_{F104A-CBD-PIP2(-)-I}$ remains in a closed state with no PIP₂ bound while KCNQ2-CaM$_{F104A-CBD-PIP2-II}$ is in an open state with the PIP₂ bound (Fig. 4c–e). The number ratio of particles used for the final reconstruction of KCNQ2-CaM$_{F104A-CBD-PIP2(-)-I}$ and KCNQ2-CaM$_{F104A-CBD-PIP2-II}$ is ~4: 1 (Fig. 4a). In comparison, only the open-state 3D reconstruction KCNQ2-CaM$_{PIP2-CBD}$ was obtained from the WT sample with both PIP₂ and CBD bound. The differential distribution of

the KCNQ2-CaM$_{F104A-CBD-PIP2}$ particles with only CBD$_A$ bound in closed and open states indicates that with the presence of PIP$_2$, (i) CBD$_A$ alone is able to stabilize some KCNQ2 channels in the open conformation, which accounts for ~20% of total channels, and (ii) CBD$_B$ further enhances this stabilization and dramatically increases the ratio of the open-conformation channels. Therefore, both CBD$_A$ and CBD$_B$ contribute to the potentiation of KCNQ2 voltage activation.

### Structural basis for the HN37 recognition

Previously we developed an effective KCNQ2 activator named HN37, which is now under clinical trial in phase I as a promising anti-epileptic drug candidate. Compared with RTG, HN37 has several advantages, such as higher activation potency to neuronal Kv7 channels, improved chemical stability, and better safety margin[34]. To reveal the activation mechanism of HN37 on KCNQ2, we then determined the HN37-bound KCNQ2 structure (KCNQ2-CaM$_{HN37}$) at 2.5 Å resolution (Fig. 5a, Table 1, Supplementary Fig. 6). In KCNQ2-CaM$_{HN37}$, to our surprise, there are also two HN37 molecules bound in each KCNQ2 subunit (Fig. 5a, b). One HN37 molecule (HN37$_A$) binds in the same pocket with RTG and CBD$_A$. HN37$_A$ mainly forms hydrophobic interactions with residues Leu299, Ile300 and Phe304 in S6$_I$, Trp236 and Phe240 in S5$_{II}$, Phe305, Pro308 and Leu312 in S6$_{II}$, and Phe104 in S1$_{III}$ (Fig. 5c). In addition, HN37$_A$ forms two hydrogen bonds with residues Trp236 and Leu299, as well as the π-π interaction with the Trp236 side chain. The other HN37 molecule (HN37$_B$) is located at the interface between VSD and PD and switched by S1$_{III}$/S4$_{III}$ and S5$_{II}$ (Fig. 5d). HN37$_B$ also mainly forms hydrophobic interactions with surrounding residues such as Val233, Trp236, and Tyr237 in S5$_{II}$, Phe100 in S1$_{III}$, and Met208 and Met211 in S4$_{III}$. Besides, one π-π interaction is observed between the fluorophenyl group of HN37$_A$ and HN37$_B$ (Fig. 5d).

To validate the two HN37 binding modes revealed by the structure data, we then introduced mutations at the two HN37 binding sites in KCNQ2 and recorded the currents of CHO cells expressing WT and mutant KCNQ2 using the whole-cell patch clamp. Under −10 mV, HN37 increases the current amplitudes of WT KCNQ2 in a dose-dependent manner with an EC$_{50}$ value of 28.52 ± 2.99 nM (Fig. 5e). In particular, 100 nM HN37 increases the current of KCNQ2 by an average factor of 2.74 ± 0.10. Meanwhile, HN37 enhances the voltage sensitivity of KCNQ2 by left-shifting the G−V curve, with an EC$_{50}$ of 27.97 ± 5.32 nM fitted from data points of $\Delta V_{1/2}$ (Fig. 5f). Specifically, 100 nM HN37 induces a shift of half maximal activation voltage ($V_{1/2}$) by −23.49 ± 0.91 mV. Among the six mutants we tested, V233A was nonfunctional. The mutation on Trp236 which interacts with both HN37 molecules, abolished the sensitivity of KCNQ2 to HN37 in both current amplitude and voltage activation (Fig. 5g, h, and Supplementary Fig. 7). For the HN37$_A$-specific interacting residues, mutation on Leu299 significantly decreased the potentiation activity of HN37 by both reducing the increase of outward current amplitude and attenuating the left-shift of the G−V curve ($\Delta V_{1/2}$) whereas mutation on Phe240 only attenuated the left-shift of the G−V curve (Fig. 5g, h). For the HN37$_B$-specific interacting residues, mutation on Met211 attenuated the left-shift of the G−V curve but did not change the current amplitude; in contrast, mutation on Phe100 did not alter HN37 sensitivity in either current amplitude or voltage activation (Fig. 5g, h). These data suggest that HN37$_A$ is more potent than HN37$_B$ in the activation of KCNQ2.

Like CBD, the HN37 alone does not allow us to capture an open-state channel. KCNQ2-CaM$_{HN37}$ is also in decoupled conformation, with activated VSDs and a closed gate (Fig. 5i, j). Although the two HN37 molecules induce a 1−2 Å shift of the VSD (Fig. 5k), the activation gate remains unchanged, with the shortest diagonal atom-to-atom distance of 4−6 Å at the activation gate (Fig. 5j).

### The HN37 and PIP$_2$ activation mechanism

To reveal the HN37 activation mechanism, we determined two KCNQ2-CaM structures in the presence of both HN37 and PIP$_2$, one with HN37 added to the protein sample first (KCNQ2-CaM$_{HN37-PIP2(-)}$, Table 1, Supplementary Fig. 8a−h) and the other one with PIP$_2$ added first (KCNQ2-CaM$_{PIP2-HN37}$, Table 1, Supplementary Fig. 9). In the map of KCNQ2-CaM$_{HN37-PIP2(-)}$, no PIP$_2$ molecule was assigned, and consequently, the KCNQ2-CaM$_{HN37-PIP2(-)}$ structure is in a closed state with two HN37 bound (Supplementary Fig. 8i, j), similar to the KCNQ2-CaM$_{HN37}$ structure (Fig. 5b). In contrast, the KCNQ2-CaM$_{PIP2-HN37}$ is in an open state with one HN37 (HN37$_A$) and one PIP$_2$ bound (Fig. 6a−c). In KCNQ2-CaM$_{PIP2-HN37}$, PIP$_2$ binds in the cleft between VSD and PD and directly interacts with Lys327 in S6 (Fig. 6d), similar to the observations in KCNQ2-CaM$_{CBD-PIP2}$ (Fig. 2b).

To analyze why HN37$_B$ does not bind in the structure of KCNQ2-CaM$_{PIP2-HN37}$, we compare structures of KCNQ2-CaM$_{PIP2-HN37}$ and KCNQ2-CaM$_{HN37}$ (Fig. 6e). Interestingly, HN37$_B$ and the hydrophobic tails of PIP$_2$ occupy the same site in KCNQ2 (Fig. 6e). With the presence of PIP$_2$ in KCNQ2-CaM$_{PIP2-HN37}$, HN37$_B$ was unable to bind. Because PIP$_2$ is essential for the activation of KCNQ channels, the clash of HN37$_B$ and PIP$_2$ in KCNQ2 suggests that HN37$_B$ may be only observed without PIP$_2$ in vitro and not exist in vivo due to the presence of PIP$_2$ in the membrane. However, we could not rule out the possibility that the two-HN37-bound KCNQ structures (KCNQ2-CaM$_{HN37}$ and KCNQ2-CaM$_{HN37-PIP2(-)}$) may present an in vivo transient state after which HN37$_B$ will be replaced by PIP$_2$ easily.

HN37$_A$ and CBD$_A$ bind in the same classical fenestration pocket and likely share similar activation mechanisms on KCNQ2. Like CBD, the binding of HN37 induces a 1−2 Å shift of the VSD (Fig. 6f), which would likely favor the binding of PIP$_2$ in vivo[45]. The binding of PIP$_2$ further destroys interactions between VSD and CaM and causes structural rearrangements of S6, CTD, and CaM, along with the opening of the activation gate (Fig. 6g). Then what is the role of HN37$_B$ if it does exist in vivo? First, by forming π-π interaction, HN37$_B$ stabilizes the binding of HN37$_A$. Second, HN37$_B$ inserts into the cleft between PD and VSD and directly interacts with S4. Thus, the HN37$_B$ likely carries part of PIP$_2$'s role by stabilizing the activated VSD before PIP$_2$ binds (Fig. 6h).

## Discussion

While here we use the homo-tetramer KCNQ2 as a model to probe the activation mechanisms of CBD and HN37, KCNQ2 and KCNQ3 can form hetero-tetrameric channels in vivo. To test whether the homo-tetramer KCNQ2 and hetero-tetrameric KCNQ2/KCNQ3 share similar ligand activations, the activation effects of 10 µM CBD and 100 nM HN37 on the activation efficacy ($I_{Drug}$ / $I_{Control}$) and the half maximal activation voltage shift ($\Delta V_{1/2}$) of wild type KCNQ2/KCNQ3 and KCNQ2(W236L)/KCNQ3 hetero-tetramers were separately examined. The sensitivity of the wild type KCNQ2/KCNQ3 to CBD and HN37 were similar to that of the KCNQ2 homo-tetramer (Figs. 1e−g and 5e, f, Supplementary Fig. 10). Consistent to the key role of Trp236 in the binding of CBD and HN37, the activation effects of the two molecules on the KCNQ2(W236L)/KCNQ3 hetero-tetramers were dramatically decreased compared to the wild type (Supplementary Fig. 10). Thus, the binding sites of CBD and HN37 are conserved among KCNQ2 and KCNQ2/KCNQ3.

Comparisons of the binding sites of RTG, CBD$_A$, and HN37$_A$ in KCNQ2 reveal a conserved agonist binding pocket in KCNQ channels, which is located in a side fenestration in TMD and lined by S5, S6, pore helix, and S6 from the adjacent subunit. While ligands bound in this pocket mainly form hydrophobic interactions with surrounding residues, specific hydrogen bonds are observed between ligands and main-chain carbonyls or side chains of nearby residues, such as Trp236, Lue299, and Ser303 (Supplementary Fig. 11). In addition, the residue Trp236, a characteristic residue for the RTG binding site in KCNQ2-5, adopts a different rotamer upon the binding of RTG and HN37 and forms π-π stacking with them. Interestingly, in the CBD-bound KCNQ2 structures, the Trp236 side chain adopts a third rotamer with no π-π stacking involved (Supplementary Fig. 11).

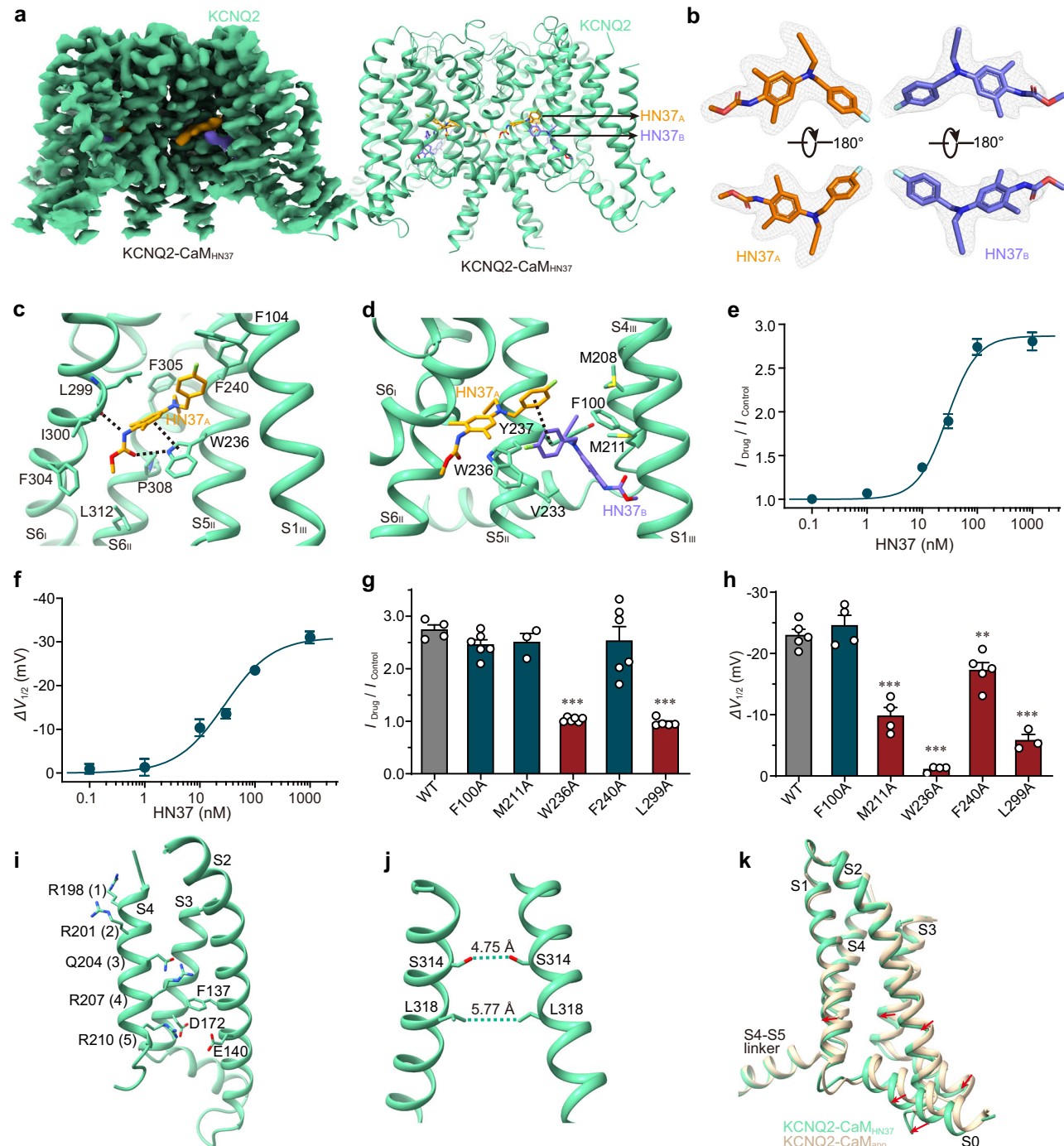

**Fig. 5 | The HN37-bound KCNQ2-CaM structure (KCNQ2-CaM$_{HN37}$). a** The 3D reconstruction and the cartoon model of KCNQ2-CaM$_{HN37}$. **b** The density maps of two HN37 molecules in different orientations at the contour level of 4.5 σ. **c** Interactions between HN37$_A$ and KCNQ2. The dashed lines indicate hydrogen bonds and π-π interaction. **d** Interactions between CBD$_B$ and KCNQ2. The dashed lines indicate π-π interaction. **e, f** Dose-response curve of the activation efficacy ($I_{Drug} / I_{Control}$) and the half maximal activation voltage shift ($\Delta V_{1/2}$) for HN37 on KCNQ2 channels. The EC$_{50}$ values were 28.52 ± 2.99 nM and 27.97 ± 5.32 nM, respectively. Data are presented as mean ± SEM. For $I_{drug} / I_{Control}$, $n = 3$ (0.1 μM), 5 (1 μM), 4 (10 μM), 4 (30 μM), 4 (100 μM), and 4 (1000 μM) individual cells; for $\Delta V_{1/2}$, $n = 3$ (0.1 μM), 3 (1 μM), 5 (10 μM), 3 (30 μM), 3 (100 μM), and 5 (1000 μM) individual cells. **g** The activation effects of 100 nM HN37 on KCNQ2 WT and mutants. Data are presented as mean ± SEM. An unpaired two-tailed $t$ test was performed. $p$ and $n$ values compared to WT ($n = 6$) are $p = 0.0727$ and $n = 6$ for F100A, $p = 0.2335$ and $n = 3$ for M211A, ***$p = 0.0001$ and $n = 6$ for W236A, $p = 0.5599$, $n = 6$ for F240A, and

***$p = 0.0001$ and $n = 5$ for L299A. $n$ represents the number of experiments from individual cells. **h** The $\Delta V_{1/2}$ of KCNQ2 WT and mutants caused by 100 nM HN37. Data are presented as mean ± SEM. An unpaired two-tailed $t$ test was used. $p$ and $n$ values compared to WT ($n = 5$) are $p = 0.4114$ and $n = 4$ for F100A, ***$p = 0.0001$ and $n = 4$ for M211A, ***$p = 0.0001$ and $n = 4$ for W236A, **$p = 0.0065$ and $n = 5$ for F240A, and ***$p = 0.0001$ and $n = 3$ for L299A. n represents the number of experiments from individual cells. **i** The activated VSD of KCNQ2-CaM$_{HN37}$. For clarity purpose, S1 is not shown. Side chains of positively charged (or polar) residues in S4 and residues forming gating charge transfer center are displayed as sticks. **j** The closed activation gate of KCNQ2-CaM$_{HN37}$. The dashed lines show diagonal atom-to-atom distance (in Å) at the constriction-lining residues Ser314 and Leu318. **k** Structural comparison of VSDs from KCNQ2-CaM$_{apo}$ and KCNQ2-CaM$_{HN37}$ when the whole channels are aligned. For (**e**–**h**), source data are included in the Source Data file.

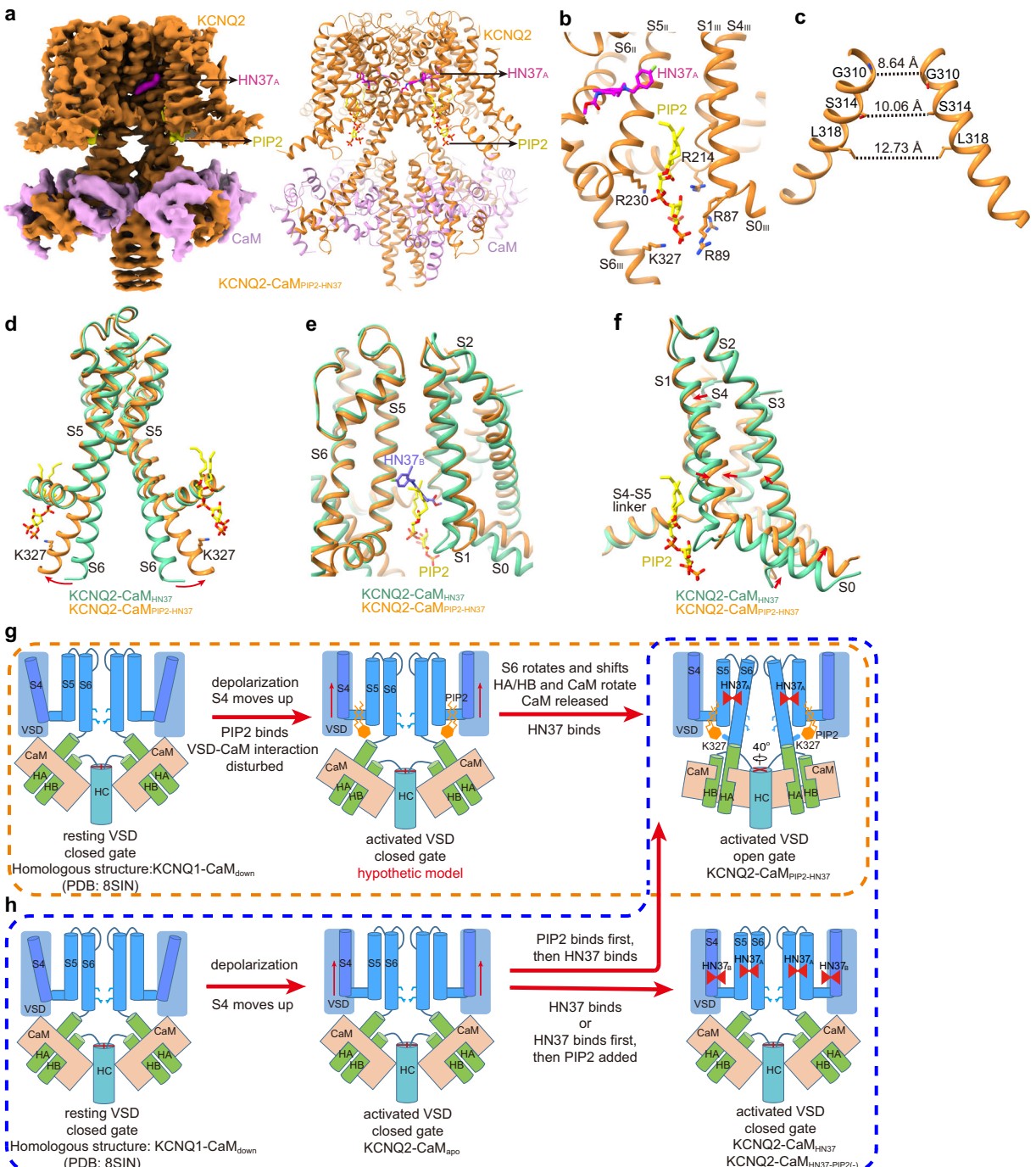

**Fig. 6 | The potential activation mechanism of HN37. a** The 3D reconstruction and the cartoon model of KCNQ2-CaM$_{PIP2-HN37}$. **b** The binding site of PIP$_2$ and HN37$_A$ in KCNQ2. Side chains of residues in KCNQ2 involved in the interactions with PIP$_2$ are shown as sticks. **c** The open activation gate of KCNQ2-CaM$_{PIP2-HN37}$. The dashed lines show diagonal atom-to-atom distance (in Å) at the constriction-lining residues Gly310, Ser314, and Leu318. **d** Overlay of the pore domains of KCNQ2-CaM$_{HN37}$ (green) and KCNQ2-CaM$_{PIP2-HN37}$ (orange) structures showing the conformational change in the ion-conducting pathway. The S6 helix bends outward upon PIP$_2$ binding at the point of PAG segment. **e** HN37$_B$ and the hydrophobic tails of PIP$_2$ occupy the same site in KCNQ2 by overlay of KCNQ2-CaM$_{PIP2-HN37}$ (orange) and KCNQ2-CaM$_{HN37}$ (green). **f** Structural comparison of VSDs from KCNQ2-CaM$_{PIP2-HN37}$ and KCNQ2-CaM$_{HN37}$ (green) when the whole channels are aligned. **g** The proposed HN37 and PIP$_2$ activation mechanism of KCNQ2 in vivo. **h** The proposed HN37 and PIP$_2$ activation model of KCNQ2 in vitro.

While CBD$_A$ and HN37$_A$ bind in the same pocket, it is interesting to speculate whether HN37$_B$ and CBD can simultaneously bind to KCNQ2. We first align structures of KCNQ2-CaM$_{HN37}$ and KCNQ2-CaM$_{CBD}$, which shows that HN37$_B$ clashes with CBD$_A$ and Trp236 side chain, but not CBD$_B$. Thus, this HN37$_B$ is unlikely to co-bind with two CBDs (Supplementary Fig. 12a). To functionally verify whether CBD and HN37 can simultaneously bind to KCNQ2, the activation effects of HN37 were tested alone or in combination with 30 μM CBD, a saturated concentration according to Fig. 1f. We found that 30 μM CBD potentiated the KCNQ2 current amplitude measured at −10 mV by 1.35 ± 0.08 folds and addition of 1 μM HN37 conferred a further increase (Supplementary Fig. 12b). The EC$_{50}$ values obtained from the dose-response curves of HN37 in the absence and presence of 30 μM CBD were 28.75 ± 6.76 nM and 178.41 ± 46.11 nM, respectively

(Supplementary Fig. 12d). The Hill coefficient was slightly changed, from 1.42 (without CBD) to 1.13 (with CBD). These data showed that CBD and HN37 exert their modulations on KCNQ2 competitively. Moreover, the maximal activation efficacy ($I_{Drug} / I_{Control}$) and the half-maximal activation voltage shift ($\Delta V_{1/2}$) of saturated HN37 + CBD on KCNQ2 were comparable to those of HN37 alone (Supplementary Fig. 12c, e, f). The data suggested that the activation effect of saturated CBD + HN37 is dominated by HN37. Taken together, these structural and electrophysiological data suggest that HN37 and CBD can not simultaneously bind to KCNQ2.

Although PIP$_2$ is essential for the activation of KCNQ channels, it is difficult to capture the PIP$_2$-bound structures of KCNQ channels with PIP$_2$ added alone in the protein sample. So far, all the PIP$_2$-bound KCNQ structures were determined in the presence of either the auxiliary subunit KCNE[41] or exogenous ligands such as ML277[39], ML213[43], CBD (Fig. 2), and HN37 (Fig. 6), suggesting low binding affinity of PIP$_2$ bound to KCNQs in vitro[47]. Structural alignments of KCNQs in the apo state and KCNE3-bound or these exogenous ligand-bound states reveals a possible consensus potentiation mechanism. KCNE3 and all these exogenous ligands induce lateral shifts of VSD relative to PD, which likely favor the binding of PIP$_2$ (Supplementary Fig. 13), although their binding sites are different among KCNQ channels.

In summary, in this study, we present structures of KCNQ2 in complex with two exogenous activators, namely CBD and HN37, with or without the endogenous lipid PIP$_2$, in either closed or open state. These structures, along with electrophysiological analyses, reveal conserved ligand activation mechanisms of KCNQ2. Both HN37 and CBD bind one KCNQ2 subunit in a 2:1 ratio – HN37$_A$ and CBD$_A$ molecules bind in the same classical fenestration pocket and HN37$_B$ and CBD$_B$ molecules in different nearby pockets. The bindings of HN37$_A$ and CBD$_A$ induce disturbance of VSD and likely favor the binding of PIP$_2$ in vivo, which transfers the conformational change of S4 upon the depolarization of the membrane potential to the CTD/CaM and S6, leading to the opening of the activation gate. The B molecules of both ligands likely stabilize the bindings of A molecules and enhance the potentiation effect. The two binding sites of both CBD and HN37 not only explain their high activation potency to KCNQ2 compared with RTG which has only one site in KCNQ2[38], but also provides clues for the development KCNQ2 activators with higher activation potency.

Previously, Mackinnon lab determined the PIP$_2$-bound open-state human KCNQ1-CaM-KCNE3 structure[41]. Similarly, we determined the activator ML277- and PIP$_2$-bound open-state structures of KCNQ1-CaM[39]. In the open-state KCNQ1, PIP$_2$ binds to S0 and the S2–S3 linker and does not directly interact with S6, suggesting that PIP$_2$ may only involve in disrupting interactions of CaM and VSD. Here the CBD- and PIP$_2$-bound open-state structure of KCNQ2-CaM reveals a different binding site and a different role of PIP$_2$. In the open-state KCNQ2, PIP$_2$ is located in the VSD and PD interface where it not only pushes VSD away from CaM but also pulls S6 up to stabilize the open-state conformation. These two different PIP$_2$ binding sites were also observed in the same channel of KCNQ4[43], suggesting the conservation and divergence of the PIP$_2$'s role in the E−M coupling of KCNQ channels.

## Methods

### Protein expression and purification
DNAs encoding human KCNQ2 (Homo sapiens: NP_004509.2) and CaM (Homo sapiens: NP_001734.1) were synthesized by GenScript. To improve the biochemical and thermal stability of KCNQ2, the N-terminal and C-terminal regions were truncated, yielding a construct containing residues 64–674. NheI and XhoI sites were used for cloning the KCNQ2 construct into the modified pEZT-BM vector with a C-terminal strep tag. The human *CaM* gene was cloned into pEZT-BM vector with a C-terminal Histidine tag. KCNQ2 and CaM complex were heterologously expressed in Human Embryonic Kidney (HEK) 293S

suspension cells (Life Technologies, ATCC, #CRL-3022) maintained at 30 °C in SMM 293-TI complete medium (Sino Biological Inc.) supplemented with 2% fetal bovine serum (FBS, Yeasen Biotechnology (Shanghai) Co., Ltd.). The P2 baculovirus was generated via the Bac-Mam system (Thermo Fisher Scientific) and used for expression when cell density reached $3.5 \times 10^6$ cells/mL. P2 baculovirus mixture of KCNQ2: CaM (6:1) was used for transduction of HEK293S cells for protein expression. To boost protein expression, 10 mM sodium butyrate was added 12 h post-transduction. Cells were harvested after 48 h, then flash-frozen in liquid nitrogen and stored at −80 °C until needed.

Cells were resuspended and lysed by sonication in buffer A (20 mM Tris, pH 8.0, 150 mM KCl) supplemented with a protease inhibitor cocktail (2 μg/mL DNase I, 0.5 μg/mL pepstatin, 2 μg/mL leupeptin, 1 μg/mL aprotinin, and 1 mM PMSF). The lysate was then solubilized with 1.5% n-dodecyl-β-D-maltoside (DDM, Anatrace) and 0.3% cholesteryl hemisuccinate tris salt (CHS, Anatrace) at 4 °C for 3 h. The insoluble cell fragment was removed by centrifugation at 48,000 g for 50 min at 4 °C. The supernatant was incubated with Strep-Tactin Sepharose resin (IBA) at 4 °C for 1.5 h with gentle rotation. Beads were loaded onto a gravity column and washed with buffer B (buffer A supplemented with 0.05% DDM and 0.01% CHS) for 4 CVs (column volumes), followed by washing with buffer C (buffer A supplemented with 0.02% GDN) for 16 CVs. The protein was then eluted with buffer C plus 10 mM d-Desthiobiotin (Sigma) and further purified on a Superose 6 gel filtration column (GE Healthcare) in buffer C. The peak fraction was collected and concentrated for cryo-EM sample preparation. For KCNQ2-CaM bound to CBD and/or PIP$_2$, the purified protein was incubated with 0.25 mM CBD (Sigma-Aldrich) and/or 1 mM PIP$_2$. For KCNQ2-CaM bound to HN37 and/or PIP$_2$, the purified protein was incubated with 0.15 mM HN37 and/or 1 mM PIP$_2$. HN37 was synthesized by the laboratory of Professor Fajun Nan (Shanghai Institute of Material Medica). The PIP$_2$ we used is 1,2-dioctanoyl-sn-glycero-3-phospho-(1′-myo-inositol-4′,5′-bisphosphate) (ammonium salt) purchased from Avanti.

### Cryo-EM data acquisition
For grids preparation, 3 μL concentrated protein was loaded onto glow-discharged R1.2/1.3 Quantifoil grids at 4 °C under 100% humidity. Grids were blotted for 4.5 s and plunge-frozen in liquid ethane using a Vitrobot Mark IV (FEI). Micrographs were acquired on a Titan Krios microscope (FEI) operating at a voltage of 300 kV. During one collection session (KCNQ2-CaM$_{HN37}$, Supplementary Fig. 6) micrographs were collected at a magnification of 49,310× (calibrated pixel size 1.014 Å, defocus range −1.1 μm to −1.3 μm, 40 frames, and a total dose 64 e⁻/Å²) using a Gatan K2 Summit direct electron detection camera via SerialEM[48] following standard procedure. After the microscope was upgraded with a Gatan Falcon 4 camera equipped with energy filter (Thermo Fisher Scientific) with slit width set to 10 eV, additional collection sessions (Supplementary Figs. 1, 2, 8, 9) were conducted at a magnification of 130,000× (calibrated pixel size 0.93 Å, defocus range −0.8 μm to −1.5 μm, 40 frames, and a total dose 52 e⁻/Å²) using EPU software (Thermo Fisher Scientific).

### Image processing
Motion correction and CTF parameters estimation were performed with the MotionCorr2[49] and the GCTF[50] programs, respectively. All image processing steps were carried out with RELION 3.1[51].

For KCNQ2-CaM$_{CBD}$, 2849 micrographs were collected and 1,034,485 particles were auto-picked and extracted with a binning factor of 3 for 2D classification. The following two rounds of 3D classification with 940,410 selected particles were performed using the map of human KCNQ2-CaM complex (PDB: 7CR3)[38] as a reference. After 3D classification, selected particles were combined and re-extracted to the pixel size of 0.93 Å for 3D refinement with a *C4*

symmetry and Bayesian polishing via RELION 3.1. The final resolution of the EM map by 3D reconstruction of 99,018 particles was 3.0 Å. For KCNQ2-CaM$_{PIP2(-)}$, KCNQ2-CaM$_{CBD-PIP2}$, KCNQ2-CaM$_{F104A-CBD-PIP2(-)-I}$, KCNQ2-CaM$_{F104A-CBD-PIP2-II}$, KCNQ2-CaM$_{HN37}$, KCNQ2-CaM$_{HN37-PIP2(-)}$, and KCNQ2-CaM$_{PIP2-HN37}$ dataset, data processing was performed following similar procedures (Supplementary Figs. 1, 2, 6, 8, 9).

The resolution was estimated by applying a soft mask around the protein density and the gold-standard Fourier shell correlation (FSC) = 0.143 criterion. Local resolution maps were calculated with RELION 3.1.

### Model building, refinement, and validation
De novo atomic models were built in Coot[52] based on the 2.5 Å resolution map of KCNQ2-CaM$_{HN37}$. The amino acid assignment was achieved based on the clearly defined density for bulky residues (Phe, Trp, Tyr, and Arg) and the model of KCNQ2-CaM complex (PDB: 7CR3)[38] was used as a reference. PHENIX[53] was utilized for model refinement against cryo-EM maps using real-space refinement, with secondary structure restraints and non-crystallography symmetry applied. The models of KCNQ2-CaM$_{PIP2(-)}$, KCNQ2-CaM$_{CBD}$, KCNQ2-CaM$_{CBD-PIP2}$, KCNQ2-CaM$_{F104A-CBD-PIP2(-)-I}$, KCNQ2-CaM$_{F104A-CBD-PIP2-II}$, KCNQ2-CaM$_{HN37-PIP2(-)}$, and KCNQ2-CaM$_{PIP2-HN37}$ were built using the model of KCNQ2-CaM$_{HN37}$ as a template. The MolProbity[54] was used for model geometry statistics generation (Supplementary Table 1). Structural figures were prepared in PyMoL (The PyMOL Molecular Graphics System, Version 1.8 Schrödinger, LLC.) and UCSF Chimera X[55].

### Mutagenesis
The wild-type human KCNQ2 cDNAs were subcloned into pcDNA5/FRT vector. Mutations were introduced into the WT cDNA constructs using the QuickChange II XL site-directed mutagenesis kit from Agilent Technologies, according to the manufacturer's manual. The mutants were verified by fully sequencing before use. Sequences of primers for mutagenesis study have been provided in the separate Supplementary Data 1.

### Cell culture and transfection
Chinese hamster ovary (CHO) cells (CCL-61) were grown in 50/50 DMEM/F-12 (Gibco) supplemented with 10% fetal bovine serum (FBS) and 2 mM L-glutamine (Life Technology). To transiently express the isoforms for electrophysiological studies, cells were seeded into 6-well plate and then transfected with 3.6 µg of the cDNA using the Lipofectamine 2000 reagent (Invitrogen) according to the manufacturer's guideline. A GFP cDNA construct (0.4 µg, Amaxa) was co-transfected to aid identification of transfected cells by fluorescence microscopy. For electrophysiological recordings, the cells were plated onto glass coverslips coated with poly-D-lysine and cultured in wells of sterile 6-well tissue culture plates in a humidified incubator at 37 °C, 5% CO$_2$, until use.

### Electrophysiological recording
Standard whole-cell voltage-clamp recording was conducted at room temperature with the Axopatch-700B amplifier. Pipettes were pulled from borosilicate glass capillaries (World Precision Instruments) with tip resistances of 3–5 MΩ when filled with the intracellular solution. The intracellular solution contained (in mM): 145 KCl, 1 MgCl$_2$, 5 EGTA, 10 HEPES (pH 7.3 adjusted by KOH); bath or extracellular solution contained (in mM): 140 NaCl, 5 KCl, 2 CaCl$_2$, 1 MgCl$_2$, 10 HEPES, and 10 glucose (pH 7.3 adjusted by NaOH). The data were filtered at 2 kHz and digitized using a DigiData 1440 A with pClamp 10.3 software (Molecular Devices). Series resistance compensation was used and set to 60%. To elicit currents, cells were stimulated by a series of 1500 ms depolarizing steps from −90 mV to +70 mV in 10 mV increments at a holding potential of −80 mV (testing CBD) or −100 mV (testing HN37). During the recordings, the bath solution was continuously perfused by a BPS perfusion system (ALA Scientific Instruments).

### Data analysis
Patch-clamp data were processed using Clampfit 10.3 (Molecular Devices, Sunnyvale, CA) and then analyzed using GraphPad Prism 5 (GraphPad Software, San Diego, CA). Voltage dependent activation curves were fitted with the Boltzmann equation, $G = G_{min} + (G_{max} − G_{min})/(1 + \exp [V−V_{1/2}]/S)$, where $G_{max}$ is the maximum conductance, $G_{min}$ is the minimum conductance, $V_{1/2}$ is the voltage for reaching 50% of maximum conductance, and S is the slope factor. Dose-response curves were fitted with the Hill equation, Y = Bottom + (Top − Bottom)/(1 + 10 ^ ((LogEC$_{50}$ − X) * P)), where X is log of drug concentration, Y is the response, Top and Bottom are corresponding to the maximum and minimum of the responses, EC$_{50}$ is the drug concentration producing the half-maximum response, and $P$ is the Hill coefficient. The data are presented as mean ± SEM. The significance was estimated using unpaired two-tailed Student's $t$ test or one-way ANOVA with Tukey's post hoc procedure. Statistical significance: *$p ≤ 0.05$, **$p ≤ 0.01$, ***$p ≤ 0.001$.

### Reporting summary
Further information on research design is available in the Nature Portfolio Reporting Summary linked to this article.

## Data availability
The data that support this study are available from the corresponding authors upon request. The cryo-EM maps have been deposited in the Electron Microscopy Data Bank (EMDB) under accession codes EMD-35884 (KCNQ2-CaM$_{PIP2(-)}$), EMD-35879 (KCNQ2-CaM$_{CBD}$), EMD-35880 (KCNQ2-CaM$_{CBD-PIP2}$), EMD-35882 (KCNQ2-CaM$_{F104A-CBD-PIP2(-)-I}$), EMD-35881 (KCNQ2-CaM$_{F104A-CBD-PIP2-II}$), EMD-35877 (KCNQ2-CaM$_{HN37}$), EMD-35883 (KCNQ2-CaM$_{HN37-PIP2(-)}$), EMD-37270 (KCNQ2-CaM$_{PIP2-HN37}$). The coordinates have been in the Protein Data Bank (PDB) under accession codes 8J05 (KCNQ2-CaM$_{PIP2(-)}$), 8J00 (KCNQ2-CaM$_{CBD}$), 8J01 (KCNQ2-CaM$_{CBD-PIP2}$), 8J03 (KCNQ2-CaM$_{F104A-CBD-PIP2(-)-I}$), 8J02 (KCNQ2-CaM$_{F104A-CBD-PIP2-II}$), 8IZY (KCNQ2-CaM$_{HN37}$), 8J04 (KCNQ2-CaM$_{HN37-PIP2(-)}$), 8W4U (KCNQ2-CaM$_{PIP2-HN37}$). Additional datasets used in this study include Protein Data Bank accession codes 7CR3, 8SIN, 6UZZ, 6V00, 7XNI, 7XNK, 7BYL, and 7VNQ. The source data underlying Figs. 1f–h, 2c, d, 5e–h, and Supplementary Figs. 1a, 2a, j, 4a–e, 6a, 7a –k, 8a, 9a, 10 b, c, 12c–f are provided as a Source Data file. Source data are provided with this paper.

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

## Acknowledgements

Single-particle cryo-electron microscopy data were collected at Center of Cryo-Electron Microscopy at Zhejiang University. We thank Dr. Xing Zhang and Dr. Shenghai Chang for support in facility access and data acquisition. This work was supported in part by the National Key Research and Development Program of China (2020YFA0908501 and 2018YFA0508100 to J.G.), Zhejiang Provincial Natural Science Foundation (LR19C050002 to J.G.), the National Science Fund for Distinguished Young Scholars (81825021 to Z.G.), National Science and Technology Innovation 2030 Major Program (2021ZD0200900 to Z.G.), the National Natural Science Foundation of China (92169202 to Z.G.), and the Youth Innovation Promotion Association of the Chinese Academy of Sciences (2020284 to Y.Z.). J.G. is supported by MOE Frontier Science Center for Brain Science & Brain-Machine Integration, Zhejiang University and Fundamental Research Funds for the Central Universities.

## Author contributions

J.G. and Z.G. conceived and supervised the project. D.M., X.L., Y.Zhang., Z.Y. and N.S. performed sample preparation, data acquisition, and structure determination. J.G., Z.G., W.Y., N.S., F.N., P.H., J.F. and G.Z. performed structure data analysis. Y.Zheng, X.Z., L.W. and W.Z. performed electrophysiological studies. All authors participated in the manuscript preparation.

## Competing interests

The authors declare no competing interests.
