## [Peer Review File · Nature Communications]

Reviewers' Comments:

Reviewer #1:

Remarks to the Author:

The voltage-gated potassium channel KCNQ2 generates the neuronal M current and plays an important role in the maintaining the stability of the membrane potential. Mutations in KCNQ2 can cause epilepsy. Understanding the structural basis of the activation of KCNQ2 by different ligands including endogenous lipids such as PIP2 and exogenous antiepileptic drugs represents a key step in the molecular mechanisms of KCNQ2 channel. In this manuscript, Ma D. et al reports cryo-EM structures of KCNQ2 in complex with PIP2, cannabidiol (CBD), and HN37, captured both closed- and opened- state structures of KCNQ2, and elucidated the ligand activation mechanisms of KCNQ2. Overall this is an important study of KCNQ2 and will guide the development of analgesics and antiepileptic drugs. Therefore, I would like to recommend minor revisions before the manuscript can be considered for publication.

1. KCNQ2 and KCNQ3 form hetero-tetrameric channels in vivo. They authors only test the ligand activation on KCNQ2 homo-tetramer, but not on the KCNQ2/KCNQ3 heterotetramer. I would suggest the authors test the activation effects of CBD and HN37 on the KCNQ2/KCNQ3 heterotetramer, at least test the WT and one of the most effective mutant, for example the W236 mutant of KCNQ2.
2. It is suggested that CBD also target other channels, such as Nav and TRPV channels. The authors may need to mention these when introducing CBD, which although are out of the scope of this manuscript.
3. A figure comparing the binding sites of retigabine, CBD, and HN37 should be better for the readers to follow.

Reviewer #2:

Remarks to the Author:

This study reports the structure of voltage-gated potassium channel KCNQ2 which related to various human diseases. By comparing multiple conformations of KCNQ2-CaM complex in the CBD-bound closed, HN37-bound closed, and CBD-PIP2 bound open states, the molecular basis for activating KCNQ2 with different small molecules was elucidated, and the development of KCNQ2 selective activators as potential analgesics and antiepileptic drugs will also be guided. Although the previous many studies on KCNQs structures may weaken the novelty of this study, the different mode of action for the activators in this study are interesting. Overall, the entire manuscript is well written, and most figures are clear. The paper deserves presentation in an outstanding journal and will be improved by consideration of the following points.

1. According to previous study, there are endogenous auxiliary subunits involved in the KCNQ family's channel switching, such as KCNQ1-KCNE3. Is there also an endogenous auxiliary subunit for KCNQ2? Or can its regulation only rely on exogenous small molecules?
2. It is interesting that two molecules of both CBD and HN37 are bound to the channel. From the figures, it seems that the HN37B would not clash with the two CBDs? If so, would the functional or structural analysis of CBD and HN37 be expected to see if CBD and HN37 can simultaneously bind to KCNQ2? It is recommended to compare the CBD-bound and HN37-bound structures to make the binding site(s) clearer.
3. As mentioned in the manuscript, HN37B and the hydrological details of PIP2 occur at the same site in KCNQ2. Can HN37 be modified in the future so that HN37B does not compete with PIP2, while also stabilizing the binding of HN37A? I have this concern because the author mentioned that HN37 is now under clinical trials. This structural study will help in improving this drug.
4. Another question on the HN37B is that whether the structure information could provide molecular basis for the advantages of the drug compared to RTG or CBD. Any experimental results or discussion are welcomed in the revised manuscript.
5. Why can PIP2 be resolved in KCNQ1-CaM-KCNE3 complex but not KCNQ2-CaM complex? Is this the result of a lack of KCNE3-like component? From this study, presence of CBD can stabilize PIP2. In the

referenced structures, PIP2 can be observed in the presence of ML277 (Ma et al. PNAS, 2022), ML213 (Zheng et al. Neuron, 2022). Are there any structural similarities between these drug-bound state and the KCNE3-bound state? Does it suggest these drugs mimic the modulation role of KCNE3 or they work in a different way?

6. Additionally, are the key residues that coordinate PIP2 in all KCNQs conserved, a sequence alignment should give a straightforward answer.

7. Line 513: the defocus range of KCNQ2-CaM-HN37 was $-1.1 \mu\text{m}$ to $-1.3 \mu\text{m}$, which is inconsistent with Supplementary Table S1.

RESPONSES TO THE REVIEWER COMMENTS

Reviewer #1 (Remarks to the Author):

The voltage-gated potassium channel KCNQ2 generates the neuronal M current and plays an important role in the maintaining the stability of the membrane potential. Mutations in KCNQ2 can cause epilepsy. Understanding the structural basis of the activation of KCNQ2 by different ligands including endogenous lipids such as PIP2 and exogenous antiepileptic drugs represents a key step in the molecular mechanisms of KCNQ2 channel. In this manuscript, Ma D. et al reports cryo-EM structures of KCNQ2 in complex with PIP2, cannabidiol (CBD), and HN37, captured both closed- and opened- state structures of KCNQ2, and elucidated the ligand activation mechanisms of KCNQ2. Overall this is an important study of KCNQ2 and will guide the development of analgesics and antiepileptic drugs. Therefore, I would like to recommend minor revisions before the manuscript can be considered for publication.

Response: We thank the reviewer's positive comments and constructive suggestions. We have collectively addressed all the reviewer's concerns. The following are our point-to-point responses to the reviewer's comments.

1. KCNQ2 and KCNQ3 form hetero-tetrameric channels in vivo. The authors only test the ligand activation on KCNQ2 homo-tetramer, but not on the KCNQ2/KCNQ3 heterotetramer. I would suggest the authors test the activation effects of CBD and HN37 on the KCNQ2/KCNQ3 heterotetramer, at least test the WT and one of the most effective mutants, for example, the W236 mutant of KCNQ2.

Response: We thank the reviewer's constructive suggestion. As suggested, the activation effects of 10 μ M CBD and 100 nM HN37 on the activation efficacy ($I_{Drug} / I_{Control}$) and the half-maximal activation voltage shift ($\Delta V_{1/2}$) of wild-type KCNQ2/KCNQ3 and KCNQ2(W236L)/KCNQ3 hetero-tetramers were separately examined. The sensitivity of the wild type KCNQ2/KCNQ3 to CBD and HN37 was similar to that of the KCNQ2 homo-tetramer (Figs. 1e-g and 5e-f, Supplementary Fig.9). Consistent with the key role of Trp236 in the binding of CBD and HN37, the activation effects of the two molecules on the KCNQ2(W236L)/KCNQ3 hetero-tetramers were dramatically decreased compared to the wild type (Supplementary Fig. 9). We have added these results in the first paragraph in Discussion (Line 329-340 Page 13) and Supplementary Fig. 9.

2. It is suggested that CBD also target other channels, such as Nav and TRPV channels. The authors may need to mention these when introducing CBD, which although are out of the scope of this manuscript.

Response: We thank the reviewer for the insightful comment. A description of the effects of CBD on Nav and TRPV have been added and four references have been cited in the revised manuscript (Line 69-71, Page 4).

3. A figure comparing the binding sites of retigabine, CBD, and HN37 should be better for the readers to follow.

Response: We thank the reviewer's constructive suggestions. The binding sites of retigabine, CBD, and HN37 have been compared and analyzed in the second paragraph in the Discussion (Line 341-350 Page 13-14) and in Supplementary Fig. 10, and presented as follows:

Comparisons of the binding sites of RTG, CBD_A, and HN37_A in KCNQ2 reveal a conserved agonist binding pocket in KCNQ channels, which is located in a side fenestration in TMD and lined by S5, S6, pore helix, and S6 from the adjacent subunit. While ligands bound in this pocket mainly form hydrophobic interactions with surrounding residues, specific hydrogen bonds are observed between ligands and main-chain carbonyls or side chains of nearby residues, such as Trp236, Lue299, and Ser303 (Supplementary Fig. 10). In addition, the residue Trp236, a characteristic residue for the RTG binding site in KCNQ2-5, adopts a different rotamer upon the binding of RTG and HN37 and forms π - π stacking with them. Interestingly, in the CBD-bound KCNQ2 structures, the Trp236 side chain adopts a third rotamer with no π - π stacking involved (Supplementary Fig. 10).

Reviewer #2 (Remarks to the Author):

This study reports the structure of voltage-gated potassium channel KCNQ2 which related to various human diseases. By comparing multiple conformations of KCNQ2-CaM complex in the CBD-bound closed, HN37-bound closed, and CBD-PIP2 bound open states, the molecular basis for activating KCNQ2 with different small molecules was elucidated, and the development of KCNQ2 selective activators as potential analgesics and antiepileptic drugs will also be guided. Although the previous many studies on KCNQs structures may weaken the novelty of this study, the different mode of action for the activators in this study are interesting. Overall, the entire manuscript is well written, and most figures are clear. The paper deserves presentation in an outstanding journal and will be improved by consideration of the following points.

Response: We thank the reviewer's positive comments and constructive suggestions. We have collectively addressed all the reviewer's concerns. The following are our point-to-point responses to the reviewer's comments.

1. According to previous study, there are endogenous auxiliary subunits involved in the KCNQ family's channel switching, such as KCNQ1-KCNE3. Is there also an endogenous auxiliary subunit for KCNQ2? Or can its regulation only rely on exogenous small molecules?

Response: We thank the reviewer's conceptual comment. Unlike the well-established modulation of KCNQ1 by KCNE auxiliary subunits, the modulation of KCNQ2 by endogenous auxiliary subunits is poorly understood. Among the 5 KCNE subunits, only KCNE2 was reported to accelerate the activation and deactivation of KCNQ2 while does not affect the current density¹. In addition, it was reported that Syntaxin-1A (Syn-1A) binds on the helix A of the C-terminus (HA) of KCNQ2 and exerts an inhibitory impact on the channel function², which needs to be re-visited given the tight interactions between HA and CaM observed in KCNQ2-CaM structures. Thus, in addition to small molecules, KCNQ2 can be regulated by endogenous auxiliary subunits, which await further investigations.

2. It is interesting that two molecules of both CBD and HN37 are bound to the channel. From the figures, it seems that the HN37B would not clash with the two CBDs? If so, would the functional or structural analysis of CBD and HN37 be expected to see if CBD and HN37 can simultaneously bind to KCNQ2? It is recommended to compare the CBD-bound and HN37-bound structures to make the binding site(s) clearer.

Response: We thank the reviewer's constructive suggestions.

To reveal whether HN37_B and two CBDs can simultaneously bind to KCNQ2, we first align structures of KCNQ2-CaM_{HN37} and KCNQ2-CaM_{CBD}, which shows that HN37_B clashes with CBD_A and Trp236 side chain, but not CBD_B. Thus, this HN37_B is unlikely to co-bind with two CBDs (Supplementary Fig. 11a). To functionally verify whether CBD and HN37 can simultaneously bind to KCNQ2, the activation effects of HN37 were tested alone or in combination with 30 μM CBD in a saturated concentration. We found that 10 μM HN37 potently activated KCNQ2 while 30 μM CBD did not confer a further potentiation on both the current amplitude and the half maximal activation voltage of KCNQ2 (Supplementary Fig. 11b, c). Then we changed the drug application order and examined their effects on KCNQ2. We observed that 30 μM CBD potentiated the KCNQ2 current amplitude by 1.35 ± 0.08 fold and addition of 1 μM HN37 conferred a further increase (Supplementary Fig. 11b). The data suggested that the activation effect of saturated CBD + HN37 is dominated by HN37. Notably, the EC₅₀ values obtained from the dose-response curves of HN37 in the absence and presence of 30 μM CBD were 28.75 ± 6.76 nM and 178.41 ± 46.11 nM, respectively (Supplementary Fig. 11d). The Hill coefficient was slightly changed, from 1.42 (without CBD) to 1.13 (with CBD). Moreover, the maximal activation efficacy ($I_{\text{Drug}} / I_{\text{Control}}$) and the half-maximal activation voltage shift ($\Delta V_{1/2}$) of saturated HN37 + CBD on KCNQ2 were comparable to those of HN37 alone (Supplementary Fig. 11e, f). CBD increased the EC₅₀ value of HN37 without effect on the HN37 maximal response, suggesting that CBD and HN37 exert their modulations on KCNQ2 competitively. Taken together, these structural and electrophysiological data suggest that HN37 and CBD can not simultaneously bind to KCNQ2.

We have added the above results in the third paragraph of the Discussion (Line 351-368 Page 14) and Supplementary Fig. 11 in the revision.

3. As mentioned in the manuscript, HN37_B and the hydrological details of PIP₂ occur at the same site in KCNQ2. Can HN37 be modified in the future so that HN37_B does not compete with PIP₂, while also stabilizing the binding of HN37_A? I have this concern because the author mentioned that HN37 is now under clinical trials. This structural study will help in improving this drug.

Response: We thank the reviewer's constructive suggestions. During the revision, we carefully examined the sample preparation for the structure determination of KCNQ2-CaM in the presence of both HN37 and PIP₂. When HN37 was first added in the sample and PIP₂ was added later, no PIP₂ molecule was observed in the structure of KCNQ2-CaM_{HN37-PIP2(-)} which is in a closed state with two HN37 bound (Supplementary Fig 7, Table 1). Interestingly, when PIP₂ was added first followed by HN37, the KCNQ2-CaM_{PIP2-HN37} structure is in an open state with one HN37 (HN37_A) and one PIP₂ bound (Supplementary Fig 8, Fig 6a-c). In KCNQ2-CaM_{PIP2-HN37}, PIP₂ binds in the cleft between VSD and PD and directly interacts with Lys327 in S6 (Fig 6d), similar to the observations in KCNQ2-CaM_{PIP2-CBD}. To analyze why HN37_B

does not bind in the structure of KCNQ2-CaM_{PIP2}-HN37, we compare structures of KCNQ2-CaM_{PIP2}-HN37 and KCNQ2-CaM_{HN37} (Fig 6e). Interestingly, HN37_B and the hydrophobic tails of PIP₂ occupy the same site in KCNQ2 (Fig 6e). With the presence of PIP₂ in KCNQ2-CaM_{PIP2}-HN37, HN37_B was unable to bind. Because PIP₂ is essential for the activation of KCNQ channels, the clash of HN37_B and PIP₂ in KCNQ2 suggests that HN37_B may be only observed without PIP₂ *in vitro* and not exist *in vivo* due to the presence of PIP₂ in the membrane. However, we could not rule out the possibility that the two-HN37-bound KCNQ structures (KCNQ2-CaM_{HN37} and KCNQ2-CaM_{HN37-PIP2(-)}) may present an *in vivo* transient state after which HN37_B will be replaced by PIP₂.

Based on these new observations in the structure of KCNQ2-CaM_{PIP2}-HN37, we have added the above analyses in the first two paragraphs in the section “The HN37 and PIP₂ activation mechanism” in the revision (Line 299-317 Page 12-13). As mentioned above, we are not sure whether HN37_B exists *in vivo*. The rational modification of HN37_B to avoid clashes with PIP₂ may be immature at the current stage.

4. Another question on the HN37_B is that whether the structure information could provide molecular basis for the advantages of the drug compared to RTG or CBD. Any experimental results or discussion are welcomed in the revised manuscript.

Response: We thank the reviewer’s constructive suggestions. If HN37_B does exist *in vivo*, two possible roles of HN37_B may render the high activation potency of HN37 on KCNQ2. First, by forming π - π interaction, HN37_B stabilizes the binding of HN37_A (Fig. 5d). Second, HN37_B inserts into the cleft between PD and VSD and directly interacts with S4. Thus, the HN37_B likely carries part of PIP₂’s role by stabilizing the activated VSD before PIP₂ binds (Fig 6e). We have updated the third paragraph in the section “The HN37 and PIP₂ activation mechanism” in the revision (Line 322-326 Page 13).

5. Why can PIP₂ be resolved in KCNQ1-CaM-KCNE3 complex but not KCNQ2-CaM complex? Is this the result of a lack of KCNE3-like component? From this study, presence of CBD can stabilize PIP₂. In the referenced structures, PIP₂ can be observed in the presence of ML277 (Ma et al. PNAS, 2022), ML213 (Zheng et al. Neuron, 2022). Are there any structural similarities between these drug-bound state and the KCNE3-bound state? Does it suggest these drugs mimic the modulation role of KCNE3 or they work in a different way?

Response: We thank the reviewer’s insightful comments and suggestions. Although PIP₂ is essential for the activation of KCNQ channels, it is difficult to capture the PIP₂-bound structures of KCNQ channels with PIP₂ added alone in the protein sample. So far, all the PIP₂-bound KCNQ structures were determined in the presence of either the auxiliary subunit KCNE³ or exogenous ligands such as ML277⁴, ML213⁵, CBD (Fig. 2), and HN37 (Fig. 6), suggesting

low binding affinity of PIP₂ bound to KCNQs *in vitro*. Structural alignments of KCNQs in the apo state and KCNE3-bound or these exogenous ligand-bound states reveals a possible consensus potentiation mechanism. KCNE3 and all these exogenous ligands induce lateral shifts of VSD relative to PD, which likely favor the binding of PIP₂ (Supplementary Fig. 12), although their binding sites are different among KCNQ channels. In this context, these drugs may mimic the modulation role of KCNE3. We have added the above structural analyses in the fourth paragraph in Discussion (Line 369-377 Page 14-15) and Supplementary Fig. 12 in the revision.

6. Additionally, are the key residues that coordinate PIP₂ in all KCNQs conserved, a sequence alignment should give a straightforward answer.

Response: We thank the reviewer's reminder. We added this sequence alignment of all KCNQs in Supplementary Fig. 4, which shows that the PIP₂-interacting residues are highly conserved across KCNQ2-5, but not in KCNQ1, which may explain why the PIP₂ binding site in KCNQ2 is not observed in KCNQ1. The analysis of the conservation amino acid involved in the PIP₂ recognition has been added in Line 179-181 Page 8.

7. Line 513: the defocus range of KCNQ2-CaM-HN37 was -1.1 μm to -1.3 μm, which is inconsistent with Supplementary Table S1.

Response: We thank the reviewer's reminder. We have corrected these defocus ranges in Supplementary Table S1.

References:

1. Tinel N, Diochot S, Lauritzen I, Barhanin J, Lazdunski M, Borsotto M. M-type KCNQ2-KCNQ3 potassium channels are modulated by the KCNE2 subunit. *FEBS Lett* **480**, 137-141 (2000).
2. Etzioni A, *et al.* Regulation of neuronal M-channel gating in an isoform-specific manner: functional interplay between calmodulin and syntaxin 1A. *J Neurosci* **31**, 14158-14171 (2011).
3. Sun J, MacKinnon R. Structural Basis of Human KCNQ1 Modulation and Gating. *Cell* **180**, 340-+ (2020).
4. Ma D, *et al.* Structural mechanisms for the activation of human cardiac KCNQ1 channel by electro-mechanical coupling enhancers. *Proc Natl Acad Sci U S A* **119**, e2207067119 (2022).
5. Zheng Y, *et al.* Structural insights into the lipid and ligand regulation of a human neuronal KCNQ channel. *Neuron* **110**, 237-247 e234 (2022).

Reviewers' Comments:

Reviewer #1:

Remarks to the Author:

The authors have responded to all my concerns.

Reviewer #2:

Remarks to the Author:

After reading the response to reviewers' comments, I think this manuscript has been well revised with appropriate changes in the figures, figure legends and main text according to the reviewers' comments. The authors addressed my questions by additional experiments or by adding the explanations and also addressed the major concerns from other reviewers as well. They presented structures, along with electrophysiological analyses, reveal conserved ligand activation mechanisms of KCNQ2. Because KCNQ2 is related to various human diseases, the molecular basis for activating KCNQ2 with different small molecules elucidated in this manuscript may provide clues for the development of analgesics and antiepileptic drugs. I think the important work of these authors deserves presentation in an outstanding journal and the manuscript can be considered for publication.